

# Ecological stoichiometric characteristics and influencing factors of carbon, nitrogen, and phosphorus in the leaves of *Sophora alopecuroides* L. in the Yili River Valley, Xinjiang

Yulu Zhang[1,2], Dong Cui[1,2,3], Yuhai Yang[3], Haijun Liu[1,2], Haijun Yang[1,2,4] and Yang Zhao[1,2]

[1] College of Biology and Geography Sciences, Yili Normal University, Yining City, Yili Kazak Autonomous Prefecture, China
[2] Institute of Resources and Ecology, Yili Normal University, Yining City, Yili Kazak Autonomous Prefecture, China
[3] State Key Laboratory of Desert and Oasis Ecology, Xinjiang Institute of Ecology and Geography, Chinese Academy of Sciences, Urumqi, China
[4] Ministry of Education Key Laboratory of Vegetation Ecology, Institute of Grassland Science, Northeast Normal University, Changchun, China

Corresponding author
Dong Cui, cuidongw@126.com

## ABSTRACT

**Background**. *Sophora alopecuroides* L. (*S. alopecuroides* L.) is a perennial herb widely distributed throughout Xinjiang, China. It is characterized by its rapid diffusion ability.
**Methods**. To reveal the ecological mechanism of the rapid spread of *S. alopecuroides*, and to elucidate the ecological stoichiometric characteristics of C, N, and P (and the influencing factors) in the leaves of *S. alopecuroides*, leaves were sampled from four habitats—forest, roadside, farmland, and desert—across the Yili River Valley. The variation rules of the ecological stoichiometric characteristics of C, N, and P in the leaves of *S. alopecuroides* were analyzed. Correlations between the ecological stoichiometric characteristics of leaves and environmental factors were examined using redundancy analysis (RDA).
**Results**. (1) The C, N, and P contents of *S. alopecuroides* leaves were 391.30–533.10 g/kg, 8.90–43.14 g/kg, and 0.71–2.04 g/kg, respectively, and the C/N, C/P, and N/P ratios were 10.34–4.94, 209.05–698.73, and 10.78–31.43 respectively. (2) The C content and C/P ratio of *S. alopecuroides* leaves were the highest in the desert habitat, leaf N content and N/P ratio were the highest in the forest habitat, leaf P content was the highest in the farmland habitat, and the leaf C/N ratio was the highest in the roadside habitat. (3) RDA showed that available potassium (AK) and pH were the main factors affecting the ecological stoichiometric characteristics of *S. alopecuroides* leaves in Yili Valley ($p \leq 0.05$), and these factors were positively correlated with C, N, P, and N/P, and negatively correlated with C/P and C/N. AK was the dominant factor that affected the P content of *S. alopecuroides* leaves, and appropriate reduction of K fertilizer would be conducive to restraining the spread of *S. alopecuroides*. Soil C, N, P, and K content, soil organic matter (OM), nitrate nitrogen ($NO_3^-$-N), ammonium nitrogen ($NH_4^+$-N), and AK had no significant effect on the ecological stoichiometric characteristics of leaves ($p > 0.05$).

## INTRODUCTION

Ecological stoichiometry is a comprehensive science that studies the equilibrium relationships, quantitative relationships, and biogeochemical cycles of various chemical elements in ecological processes (*Sterner & Elser, 2002*), providing an important technical means for analyzing the nutrient utilization of vegetation (*Elser & Schampel, 1996*). The essence of plant growth is the regulation of the accumulation and relative proportions of C, N, and P in plants (*Koerselman & Meuleman, 1996*). C, N, and P are essential nutrients for plant growth and development. C is the most important element in plant dry matter; N promotes the synthesis of amino acids and proteins, but also enhances the photosynthetic capacity of plants, and P is not only an important component of nucleic acids and enzymes, but also a basic element of all living organisms; thus, C, N, and P significantly affect plant growth and the regulation of physiological mechanisms (*Wang et al., 2011*).

Previous studies on the stoichiometric characteristics of C, N, and P have primarily focused on the stability of plant communities (*Tessier & Raynal, 2003*), litter decomposition processes (*Mooshammer et al., 2012*), and determination of plant growth limiting elements (*Tjoelker et al., 2005*). The study of plant leaf ecological stoichiometry helps to explore plant growth characteristics and nutrient limitations. Leaves are the main sites of plant photosynthesis, and rate of photosynthesis are closely related to the N content of the leaves. The C/N and C/P ratios in leaves reflect the rate of plant carbon assimilation, and—to a certain extent—can reflect plant nutrient use efficiency (*Wang & Yu, 2008*). Leaf N/P is a sensitivity index and an important evaluation tool for plant growth nutrient restriction (*Duan et al., 2004*). In addition, growth of plants has been found to be not only affected by C, N, and P levels in the plant tissues, but also by the external environment (*Cleland, 2011*). Soil moisture, salinity, and nutrients have a significant impact on the C, N, and P content of plant leaves and their stoichiometric ratios (*Chen et al., 2016*; *Yan et al., 2011*). The effective absorption of soil physical-chemical factors and soil nutrients will affect the ecological stoichiometric characteristics of plant leaves. Furthermore, the soil nutrient content and balance are closely related to the ecological stoichiometric characteristics. Soil nutrient levels not only affect plant growth and community composition but also indicate the health of the ecosystem (*Li et al., 2014*). Therefore, it is important to explore the characteristics of nutrient elements in the leaves and the physicochemical properties of the soil, which to understand nutrient utilization in plant growth and the adaptation mechanisms of plants to the environment.

*Sophora alopecuroides* L. (*S. alopecuroides* L.) is a perennial herb of the legume family and is predominantly distributed in arid desert and grassland marginal areas, such as Xinjiang, Ningxia, and Inner Mongolia. It is characterized by salinity tolerance and drought resistance (*Qi, He & Shi, 2008*). Therefore, it is widely used in wind barriers, sand fixation, and saline-alkali land improvement (*Chen & Jia, 2000*). Due to the fast-spreading

characteristics of *S. alopecuroides*, populations often grow continuously and are distributed widely, which can form a single-species dominant community in a short time. In recent years, the grassland in the Yili River Valley of northwestern Xinjiang has been degenerating, in association with the spread of many poisonous weeds. Among them, *S. alopecuroides* has rapidly become the dominant species due to its rapid diffusion characteristics, occupying the living space of other species, and thus contributing to the degradation of the grasslands. The dominant presence of *S. alopecuroides* poses a serious threat to the development of local animal husbandry and biodiversity (*Cui et al., 2018*). Current research on *S. alopecuroides* mainly focuses on seed morphology, medicinal value, and seed dormancy and germination conditions (*Liu et al., 2017*; *Hao, Meng & Jie, 2016*; *Wang et al., 2007*). There are few reports addressing the ecological stoichiometric characteristics of *S. alopecuroides* leaves or their correlation with soil physicochemical factors.

The objective of this study was to reveal the survival strategies and ecological mechanisms leading to the rapid spread of *S. alopecuroides* in arid and semi-arid areas, and the effects of *S. alopecuroides* on soil quality in degraded steppes, to provide a theoretical basis for the scientific management and ecological restoration of degraded grasslands. Therefore, from the perspective of ecological stoichiometric characteristics, the present study examined *S. alopecuroides* leaves from the Yili River Valley, to systematically analyze the variation of the ecological stoichiometric characteristics of leaf-tissue C, N, and P levels across various habitats. Relationships were examined between environmental factors and the ecological stoichiometric characteristics of *S. alopecuroides*.

## MATERIAL AND METHODS

### Site description

The study area was located in the Yili River Valley of the Xinjiang Uygur Autonomous Region (80°09′–84°56′E, 42°14′–44°50′N). High mountains form the north, east, and south sides of the Yili River Valley. The terrain ranges from high in the east to low in the west, in the shape of a trumpet, so the natural landform can be described as "three mountains and two valleys," and is considered a "wet island." The elevation ranges from 530 m to 1,000 m. The valley spans 360 km from east to west and 275 km from north to south, covering an area of 56,400 km². The Yili River Valley is the wettest area in Xinjiang, with a warm and humid temperate continental climate. The average annual temperature is 10.4 °C. Annual sunshine ranges between 2700 and 3000 h, and the average annual precipitation is 417.6 mm, of which approximately 60~70% occurs during the spring and summer. The Yili River Valley has abundant natural resources, high species diversity, various mineral deposits, and unique wetland landscapes. Vegetation types in the valley are mainly grassland, meadow, and forest.

### Study site and sample collection

Field investigation and observation revealed that *S. alopecuroides*, having strong invasion ability, often colonizes four landscapes—forest, roadside, farmland, and desert—in which it becomes widely distributed. Sampling sites were established in Qapqal County, Yili River

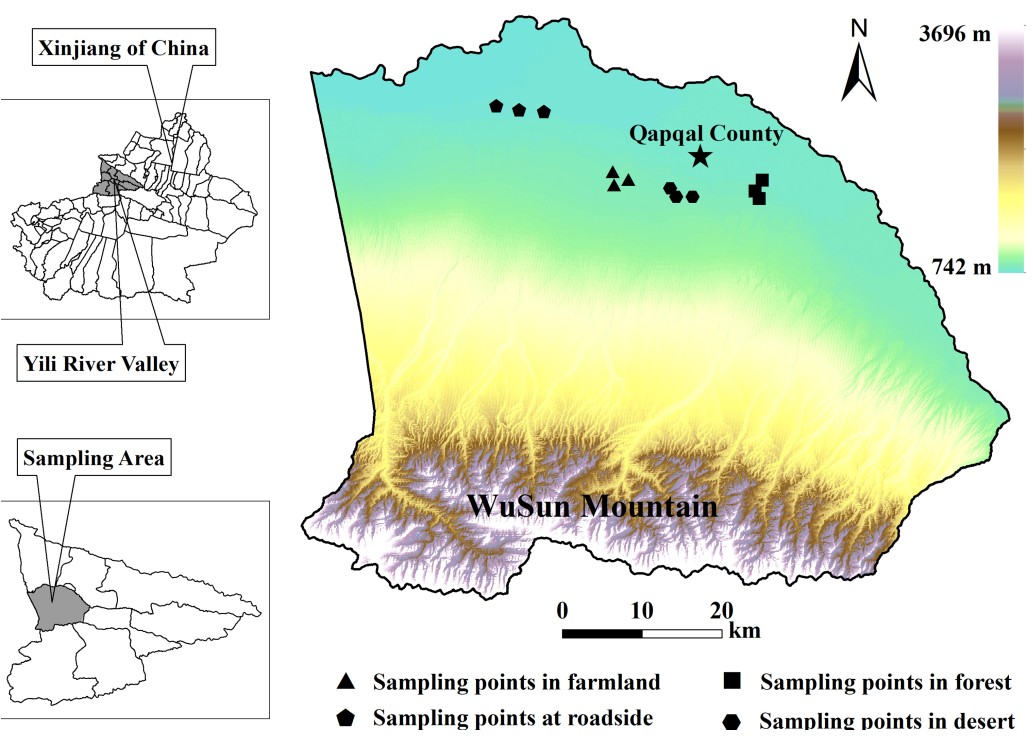

**Figure 1** Diagram of sampling points in Yili Valley.

Valley, covering those four habitat types (Fig. 1). Soil samples were collected in September 2018, and plant samples were collected in July 2019.

In the four invasive habitats of *S. alopecuroides*, 20 m × 20 m plots were selected respectively, and 3 quadrats were randomly selected from each plot, with an area of 5 m × 5 m. In each plot, 20 complete *S. alopecuroides* leaves were randomly selected, and taken back to the laboratory for cryopreservation. All collected leaves were dried at 70 °C for 24 h. The leaves were then crushed into powder using a pulverizer, then sealed for storage. The contents of total C, total N, and total P in the leaves of *S. alopecuroides* leaves were determined.

According to the same method, 20 m × 20 m plots were set in four invasive habitats of *S. alopecuroides*, and three quadrats (5 m × 5 m) were randomly set in each plot. Soil samples of 0~10 cm, 10~20 cm, and 20~30 cm depth were collected from each quadrats. 36 soil samples were collected from the four habitats. The collected soil samples were placed in sealed plastic bags. After transportation to the laboratory, the soil samples were dried, ground with a mortar and pestle, then screened through a 0.5 mm mesh for soil physical-chemical properties o analyzed.

## Analysis of soil and plant properties
### Leaf properties of plants
The total C content of the *S. alopecuroides* leaves was measured by a $K_2CrO_7$-$H_2SO_4$ oxidation procedure (*Bao, 2000*). For the total N content (TN) in the soil and plant

samples, firstly, the soil and plant samples were boiled with perchloric acid ($HClO_4$) and sulfuric acid ($H_2SO_4$), and then the total N content was determined by FOSS 1035 automatic nitrogen analyzer (*Bao, 2000*). To establish the total P content, firstly, the leaves and soil samples were boiled in perchloric acid and sulfuric acid, then the colorimetric method was applied using an Agilent CARY 60 ultraviolet–visible spectrophotometer; finally, the total phosphorus content (TP) of soil and plant samples was measured (*Bao, 2000*). The contents of C, N and P were measured in g/kg.

## Soil physicochemical properties

The total potassium (K) content in the soil was measured using atomic absorption spectrophotometry (*Bao, 2000*). To determine the content of ammonium nitrogen ($NH_4^+$-N) and nitrate nitrogen ($NO_3^-$-N) in the soil, firstly, 10.00 g of the soil sample was weighed into a plastic bottle, calcium chloride ($CaCl_2$) extractant was added, and the mixture was shaken for 30 min between 20 °C ~25 °C, then measured the content of $NH_4^+$-N and $NO_3^-$-N by colorimetry after filtration (*Bao, 2000*). The content of soil organic matter (OM) was measured by a $K_2CrO_7$-$H_2SO_4$ oxidation procedure (*Bao, 2000*). Soil pH was measured using a pH meter; To determine soil available phosphorus (AP), 2.50 g of each soil sample was weighed into a plastic bottle, then $NaHCO_3$ extract and 1 g of phosphorus-free activated carbon was added; The sample was shaken for 30 min between 20 °C ~25 °C, then filtered, and the content of AP in the soil sample was measured by colorimetry (*Bao, 2000*). The content of available potassium (AK) was determined by the flame photometric method (*Bao, 2000*).

## Statistical analysis

Excel 2010 (Microsoft) and SPSS 19.0 statistics software were used to analyze the data after integration. One-way ANOVA method was used to analysis the differences in C, N, and P contents and their stoichiometric ratios in the leaves of *S. alopecuroides* across the four different habitats. And significance analysis was performed using Duncun multiple comparison. It should be noted that the factors significantly related to the ecological stoichiometric characteristics of C, N, and P need to be selected by Monte Carlo analysis before the redundancy analysis. According to the detrended correspondence analysis (DCA) of leaf C, N, and P contents, it can be seen that the length of gradient (LGA) of the sorting axis was less than 3, indicating the data had suitable linearity between the leaf nutrient levels and the soil environmental factors, which is suitable for the redundancy analysis (RDA).

In the sorting diagram, the quadrant in which the arrow is located represents a positive and negative correlation between the factors and the sorting axis, while the hollow arrow represents the ecological stoichiometric characteristics of the leaves. The solid arrow represents the physicochemical factors of the soil. The length of the line represents the relationship between the ecological stoichiometric characteristics of the leaves of *S. alopecuroides* and the soil chemical factors. The angle between the two arrows represents the correlation between the ecological stoichiometric characteristics of the leaves and soil chemical factors. The smaller the angle, the greater is the correlation. The solid line

represents the environmental factors that were significantly related to the stoichiometric characteristics of the leaves ($p < 0.05$).

## RESULTS

### Content and stoichiometric ratios of C, N, and P in *S. alopecuroides* leaves

As indicated by Table 1, the average values of C, N, and P contents in the leaves of *S. alopecuroides* in the Yili River Valley were 470.09 g/kg, 32.71 g/kg, and 1.43 g/kg, respectively. The coefficients of variation of C, N, and P were 10.96%, 30.41%, and 30.86%, respectively. The average values of C/N, C/P, and N/P were 16.88, 364.67, and 23.20, respectively, and the variation coefficients were 57.04%, 38.42%, and 24.00%, respectively. The coefficients of variation of C, N, P, and the stoichiometric ratios of leaves were generally large, with the largest being the coefficient of variation of the C/N ratio, indicating that the C and N contents of the leaves had the highest degree of variation and the strongest variability. As shown in Figs. 2A and 2B, there was no significant correlation between leaf C content and leaf N and P content ($p > 0.05$), while there is a strong positive correlation ($p < 0.01$) between leaf N and P contents (Fig. 2C).The regression equation ($y = 0.0009x^2 - 0.0095x + 0.7532$) clearly reflects the increasing trend of P content in leaves with increasing N content.

### Contents and stoichiometric ratios of C, N, and P in *S. alopecuroides* leaves in different habitats

There were some differences in the C and P contents of *S. alopecuroides* leaves in different habitats (Table 2). The C content of leaves in the habitats showed an increasing trend: the C content of leaves from the desert habitat was much higher than those from the forest. The coefficients of variation of C content in the leaves were 11.58%, 13.28%, 14.93%, and 5.73% for the forest, roadside, farmland, and desert habitats, respectively. There was no significant difference in N content among the four habitats. The variation coefficients of N content in the forest, roadside, farmland, and desert habitats were 7.52%, 63.30%, 3.37%, and 31.24%, respectively. The P content of the leaves sampled from the four habitats in order from highest to lowest was farmland >forest >roadside >desert, and the P content of the leaves from the desert was significantly lower than that of the farmland. The variation coefficients of P content in the leaves of the forest, roadside, farmland, and desert were 13.62%, 40.12%, 6.96%, and 44.23%, respectively.

There was a significant difference in the stoichiometric ratio of C/P in the leaves of different habitats, but there were no significant differences in C/N and N/P (Table 3). The C/N ratio of the forest leaves was slightly lower than those of the desert, roadside, and farmland areas. The coefficients of variation of C/N in the forest, roadside, farmland, and desert were 7.59%, 75.31%, 12.19%, and 22.65%, respectively. The N/P ratio of leaves from the farmland habitat was less than that of the forest, roadside, and desert areas. The coefficients of variation of N/P in the forest, roadside, farmland, and desert habitats were 13.31%, 41.23%, 5.09%, and 19.31%, respectively. The leaf C/P of the four habitats, from largest to smallest, was desert >roadside >forest >farmland, and the leaf C/P of the farmland habitat
**Table 1  Ecological stoichiometry characteristics of C, N and P in the *Sophora alopecuroides* leaves.**

|  | C(g/kg) | N(g/kg) | P(g/kg) | C/N | C/P | N/P |
|---|---|---|---|---|---|---|
| Mean | 470.09 | 32.71 | 1.43 | 16.88 | 364.67 | 23.20 |
| Median | 482.26 | 36.70 | 1.53 | 13.48 | 314.75 | 23.06 |
| Standard error | 14.87 | 2.87 | 0.13 | 2.78 | 40.44 | 1.61 |
| Range | 391.30~533.10 | 8.90~43.14 | 0.71~2.04 | 10.34~44.94 | 209.05~698.73 | 10.78~31.43 |
| Coefficient of variation (%) | 10.96 | 30.41 | 30.86 | 57.04 | 38.42 | 24.00 |

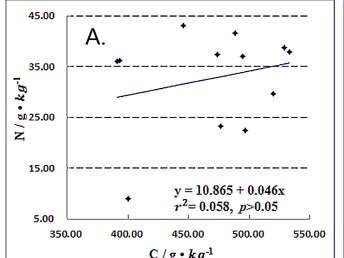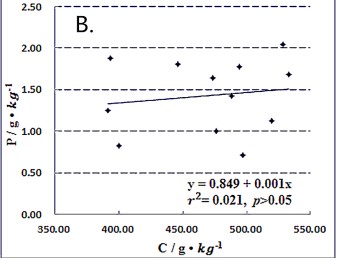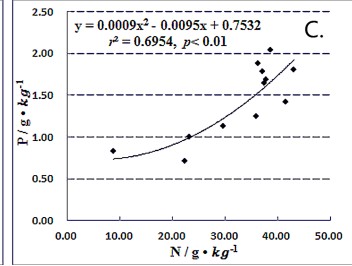

**Figure 2  Correlation of contents of C, N, and P in leaves of *S. alopecuroides*.**

**Table 2  Contents of C, N and P in leaves of *Sophora alopecuroides* in different habitats and their variation coefficients.**

|  | Forest | Roadside | Farmland | Desert |
|---|---|---|---|---|
| C (g/kg) | $(450.98 \pm 52.21)^b$ | $(455.22 \pm 60.44)^{ab}$ | $(472.05 \pm 70.47)^{ab}$ | $(502.09 \pm 28.76)^a$ |
| Coefficient of variation (%) | 11.58 | 13.28 | 14.93 | 5.73 |
| N (g/kg) | $(38.35 \pm 2.88)^a$ | $(27.25 \pm 17.25)^a$ | $(37.39 \pm 1.26)^a$ | $(27.83 \pm 8.69)^a$ |
| Coefficient of variation (%) | 7.52 | 63.30 | 3.37 | 31.24 |
| P (g/kg) | $(1.44 \pm 0.20)^{ab}$ | $(1.25 \pm 0.50)^{ab}$ | $(1.90 \pm 0.13)^a$ | $(1.13 \pm 0.50)^b$ |
| Coefficient of variation (%) | 13.62 | 40.12 | 6.96 | 44.23 |

**Notes.**
The value is (mean ± SD). Different letters in the upper right corner of the peer data indicate that the data in different habitats are significantly different ($p < 0.05$).

was significantly lower than that of the desert. The coefficients of variation of C/P in the forest, roadside, farmland, and desert habitats were 8.69%, 32.98%, 14.29%, and 38.67%, respectively.

## Correlation between ecological stoichiometric characteristics of *S. alopecuroides* leaves and soil physicochemical factors

RDA was used to study the correlation between the ecological stoichiometric characteristics of *S. alopecuroides* leaves and the soil physicochemical factors (AK, pH, $NO_3^-$-N, $NH_4^+$-N, AP, OM, soil C content, soil N content, soil P content, and soil K content). It can be seen from Table 4 that the interpretation amounts of the first and second sorting axes were 54.8% and 26.9%, respectively. The first two axes jointly explained 81.7% of the change in the leaf ecological stoichiometric characteristics. The cumulative explanations of the leaf ecological stoichiometric characteristics and soil physicochemical factors of *S. alopecuroides*

**Table 3 Ratios of carbon-nitrogen, carbon-phosphorus, nitrogen-phosphorus and coefficient of variation in leaves of *Sophora alopecuroides* in different habitats.**

|  | Forest | Roadside | Farmland | Desert |
|---|---|---|---|---|
| C/N | $(11.75 \pm 0.89)^a$ | $(24.26 \pm 18.27)^a$ | $(12.60 \pm 1.54)^a$ | $(18.93 \pm 4.29)^a$ |
| Coefficient of variation (%) | 7.59 | 75.31 | 12.19 | 22.65 |
| C/P | $(315.27 \pm 27.39)^{ab}$ | $(397.90 \pm 131.21)^{ab}$ | $(248.56 \pm 35.51)^b$ | $(496.97 \pm 192.18)^a$ |
| Coefficient of variation (%) | 8.69 | 32.98 | 14.29 | 38.67 |
| N/P | $(26.99 \pm 3.59)^a$ | $(20.37 \pm 8.40)^a$ | $(19.71 \pm 1.00)^a$ | $(25.71 \pm 4.97)^a$ |
| Coefficient of variation (%) | 13.31 | 41.23 | 5.09 | 19.31 |

Notes.
The value is (mean $\pm$ SD). Different letters in the upper right corner of the peer data indicate that the data in different habitats are significantly different ($p < 0.05$).

**Table 4 Correlation between ecological stoichiometry characteristics of leaves and sorting axis.**

| Sort axis | The axis I | The axis II | The axis III | The axis IV |
|---|---|---|---|---|
| Characteristic value | 0.548 | 0.269 | 0.113 | 0.008 |
| Correlation between leaf stoichiometric characteristics and factors of soil physical and chemical | 0.986 | 0.980 | 0.893 | 0.972 |
| Cumulative interpretation of stoichiometric characteristics (%) | 54.8 | 81.7 | 93.0 | 93.8 |
| Stoichiometric characteristics and cumulative interpretation of factors for soil physical and chemical (%) | 58.3 | 86.8 | 98.8 | 99.7 |
| Typical eigenvalues |  | 0.941 |  |  |
| Total eigenvalue |  | 1 |  |  |

leaves reached 86.8%, indicating that the first two axes could reflect the large difference between the soil physicochemical factors and the leaf stoichiometric characteristics, and were mainly determined by the first sorting axis.

According to the RDA (Fig. 3), the arrow line between AK and pH was the longest, which is consistent with the importance ranking results in Table 5. Together, AK and pH had the greatest impact on the ecological stoichiometric characteristics of *S. alopecuroides* leaves. AP and pH were positively correlated with leaf C, N, P, and N/P, and negatively correlated with leaf C/P and C/N. The direction of the arrow line of AK and leaf P content was almost the same with a small angle, indicating that AK was significantly positively related to leaf P content, and AK may be an important factor affecting leaf P content in *S. alopecuroides* leaves in the Yili River Valley.

Different soil physicochemical factors were found to exhibit significant differences in the ecological stoichiometric characteristics of *S. alopecuroides* leaves (Table 5). The effects of soil physicochemical factors on stoichiometric characteristics were as follows: AK >pH >$NO_3^-$-N >soil P content >$NH_4^+$-N >soil N content >soil K content >soil OM >soil C content >AP. AK and pH had a significant effect on the stoichiometric characteristics of the leaves ($p \leq 0.05$). AK had the most significant effect on the stoichiometric characteristics of leaves, accounting for 19.9% of the total interpretation (4.487, $p = 0.05$). $NO_3^-$-N, soil P content, $NH_4^+$-N, soil N content, soil K content, OM, soil C content, and AP had no significant effect on the stoichiometric characteristics of the leaves ($p > 0.05$).

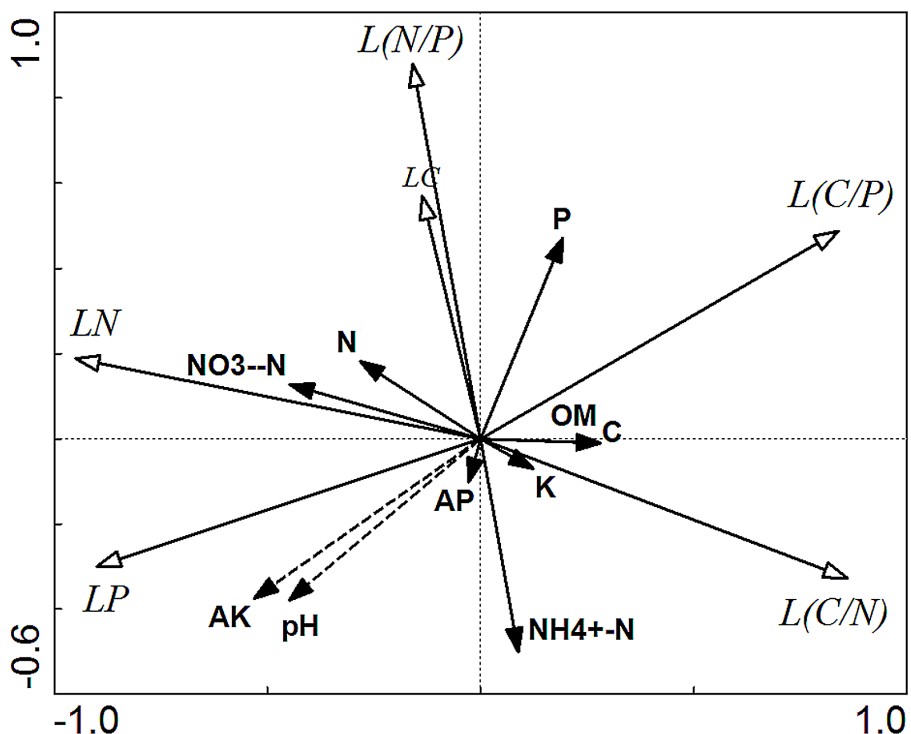

**Figure 3  Redundancy analysis of the influence of soil physicochemical properties on the ecological stoichiometric characteristics of leaves.** LC, (leaf C) carbon content of *S. alopecuroides* leaves; LN, (leaf N) nitrogen content of leaves; LP, (leaf P) phosphorus content of leaves; L(C/N), carbon–nitrogen ratio of leaves; L(N/P), nitrogen–phosphorus ratio of leaves; L(C/P), carbon–phosphorus ratio of leaves; C, soil carbon content; N, soil nitrogen content; P, soil phosphorus content; K, soil potassium content.

**Table 5  Significance rank and significance test of soil physicochemical factors in explanation.**

| Environmental factor | Significance rank | Explanatory capacity (%) | Importance ($F$ value) | Saliency ($p$ value) |
|---|---|---|---|---|
| AK | 1 | 19.9 | 4.487 | 0.042 |
| pH | 2 | 15.4 | 2.916 | 0.050 |
| $NO_3^-$-N | 3 | 14.1 | 1.646 | 0.188 |
| Soil P content | 4 | 8.6 | 0.939 | 0.418 |
| $NH_4^+$-N | 5 | 8.5 | 0.924 | 0.444 |
| Soil N content | 6 | 5.8 | 0.620 | 0.628 |
| Soil K content | 7 | 4.9 | 0.512 | 0.722 |
| OM | 8 | 4.6 | 0.480 | 0.72 |
| Soil C content | 9 | 4.5 | 0.476 | 0.698 |
| AP | 10 | 0.7 | 0.073 | 0.982 |

## DISCUSSION

### Ecological stoichiometric characteristics of C, N, and P in the leaves of *S. alopecuroides* in different habitats

The average C content (470.09 mg/g) of *S. alopecuroides* leaves in the Yili Valley was slightly higher than the average of 492 global terrestrial plants leaves (464 mg/g; *Elser et al., 2000a*), indicating that the organic C content of *S. alopecuroides* in the Yili River valley was higher, therefore the ability to store C storage was stronger. There was a certain difference in leaf C content among the four habitats, which indicated that leaf carbon accumulation of *S. alopecuroides* in the four habitats was different.

The higher C content from the desert leaves may be because carbon usually exists in plants in the form of OM. Under conditions of low soil moisture and high salt content, a habitat will likely be high in stress and low in competitive interference. *S. alopecuroides* readily stores carbon, reduces its reproductive and competitive ability, maintains normal growth, and achieves balanced resource allocation (*Zhang, 2000*). It may also be the case that in an environment with sufficient resources, it is easy to reach environmental capacity saturation, which would aggravate interspecific competition and reduces the available natural resources.

Yili Valley lies within the arid and semi-arid region of Xinjiang. The average N content in the leaves of *S. alopecuroides* (32.71 mg/g) is not only much higher than the average of Chinese plant leaves (20.2 mg/g; *Han et al., 2005*), but also higher than average N content in plant leaves at a global scale (20.6 mg/g; *Elser et al., 2000a*). This seems consistent with previous observations that the N content of leaves from arid desert environments was relatively high (*Li et al., 2010*); however, the N content in *S. alopecuroides* leaves showed no significant difference across the four habitats. *S. alopecuroides* is a leguminous plant with strong nodulation, and therefore nitrogen-fixation ability; this would explain the high N content in the leaves, as well as the high internal stability of N levels.

The average content of P in the leaves of *S. alopecuroides* (1.43 mg/g) was lower than the average of plants in China (1.5 mg/g; *Han et al., 2005*) and of that of global plants (1.99 mg/g; *Elser et al., 2000a*). The abundant precipitation of the Yili River Valley's temperate continental climate enhances the leaching of available P in the soil—which is not conducive to binding and accumulating P—resulting in less P absorption by plant leaves (*Wang et al., 2018*). Previous studies have shown that human disturbance, such as the application of fertilizer, affects the plant chemometrics. Cultivation, fertilization, irrigation, and other activities improve the local soil's nutrient content and quality and improve the effective P content (*Luo & Gong, 2016*), which would provide a favorable environment for *S. alopecuroides* growth. This explains the higher P content in the *S. alopecuroides* leaves from the farmland habitat compared to the other habitats.

C/N and C/P represent the ability to assimilate C when plants absorb nutrient elements. To some extent, it can reflect the utilization efficiency of nutrient elements in plants, and the ratios are closely related to the growth rates of organisms (*Davis, Childers & Noe, 2006*). In this study, the C/N ratio (16.88) of *S. alopecuroides* leaves was lower than the global plant average (22.5; *Elser et al., 2000b*), and C/P (364.67) was much higher in the *S. alopecuroides*
leaves than the global plant average (232; *Elser et al., 2000b*), indicating that the nutrient utilization efficiency of *S. alopecuroides* L. was low. As shown in Table 3, the C/N and C/P ratios of *S. alopecuroides* leaves grown in the desert and roadside habitats were higher than those of the farmland and forest habitats. This finding may be because, in an environment with sufficient nutrition, the growth rate of plants is high, the synthesis of organic carbon is high, and the C/N and C/P ratios are low. In nutrient-deficient environments, plant growth is relatively slow, the utilization efficiency of nutrient elements is high, and the C/N and C/P ratios are high (*Ng et al., 2014*).

The ratio of N to P in plant leaves reflects the dynamic balance between soil nutrient supply and plant nutrient demand; the N/P ratio can be used to determine the limiting growth factors of plant nutrients (*Duan et al., 2004*). Studies have shown that when leaf N/P <14, plant growth is mainly restricted by N; when leaf N/P >16, plant growth is mainly restricted by P; and when leaf NP is more than 14, but less than 16, plant growth is restricted by both N and P (*Aerts & Chapin, 2000*). *S. alopecuroides* is a nitrogen-fixing plant species, so it is generally N-limited, due to its higher leaf P content (mean = 1.53 mg/g), indicating that it may not be P-limited. Therefore, the rule-of-thumb for judging restrictive elements—that is, an N/P of 14 or 16—may not greatly apply to nitrogen-fixing plants. Further studies are needed to determine which nutrients limit plant growth.

## Factors affecting the ecological stoichiometric characteristics of C, N, and P in *S. alopecuroides* leaves

Plants need to absorb nutrients from the soil to supplement the nutrients needed for the growth and development of leaves, so soil physicochemical factors have a greater impact on the C, N, and P ecological stoichiometric characteristics of *S. alopecuroides* leaves. According to the observed RDA ranking, AK and pH were the main factors affecting the C, N, and P stoichiometric characteristics. Soil pH can change soil nutrient content and distribution area, thus affecting plant growth and developmental processes (*Zhan, Yu & He, 2013*). In this study, pH was positively correlated with C, N, and P contents and N/P in *S. alopecuroides* leaves and was negatively correlated with C/P and C/N, indicating that pH is closely related to leaf growth, which is similar to the finding that pH affects the growth and development of plants by affecting the physical, chemical, and biological characteristics of soil (*Xu et al., 2015*). In this study, it was observed that with an increase in the content of AK, the C, N, P, the N/P ratio in the leaves also increased, but C/P and C/N decreased. The content of P in the leaves was positively correlated with the content of AK, indicating that AK was the main factor affecting the content of P in the leaves. This may be because the absorption efficiency of soil nutrients to *S. alopecuroides* leaves is different in arid and semi-arid areas compared to other environments. The absorption efficiency of AK is higher in arid and semi-arid areas, which benefits the growth of *S. alopecuroides* leaves.

The internal stability of grassland ecosystems in the arid and semi-arid areas of Xinjiang was explored by analyzing the influence of soil physicochemical factors on the ecological stoichiometric characteristics of *S. alopecuroides* leaves. The RDA indicated that the stoichiometric characteristics of C, N, and P in the sampled *S. alopecuroides* leaves were not significantly affected by the OM, nitrate nitrogen, ammonium nitrogen, AK, or the soil C,

N, P, and K levels. However, the independent analysis of the effect of soil physicochemical factors on the stoichiometric characteristics showed some deficiencies. First, the effects of soil physicochemical factors on the C, N, and P ecological stoichiometric characteristics of *S. alopecuroides* leaves were not independent. Second, soil physicochemical factors had mutual influence and restrictions. Therefore, based on the independent analysis, it is necessary to further analyze the double or even multiple synergistic effects of soil physicochemical factors on the ecological stoichiometric characteristics of C, N, and P in *S. alopecuroides* leaves, to make the results more accurate (*Li et al., 2019*).

As the dominant species degrading grassland in the Yili River Valley, the growth, development, and distribution of *S. alopecuroides* seriously affects the ecosystem in the valley. Studying the relationships between the C, N, and P ecological stoichiometric characteristics of leaves and environmental factors is of great significance in revealing the ecological mechanisms of the successful diffusion of *S. alopecuroides* plants in the Yili River Valley.

## CONCLUSIONS

(a) The contents of C, N, and P in the leaves of *S. alopecuroides* in Yili Valley were 391.30–533.10 g/kg, 8.90–43.14 g/kg, and 0.71–2.04 g/kg, respectively. The C/N, C/P, and N/P ratios were 10.34–44.94, 209.05–698.73, and 10.78–31.43, respectively. Compared with the global and national plant leaf averages, the C and N contents of the *S. alopecuroides* leaves in the arid desert environment were higher.

(b) There were significant differences in C and P contents, and the C/P ratios between *S. alopecuroides* leaves in different habitats, but there were no significant differences in the C and N content or the C/N and N/P ratios in leaves from different habitats. The order of leaf P content in the four habitats was farmland >forest >roadside >desert, and the order of leaf C/P was desert >roadside >forest >farmland, indicating that reclamation of farmland could boost the content of AP and provide a favorable environment for the growth of *S. alopecuroides*.

(c) AK and pH were the main factors affecting the ecological stoichiometric characteristics of *S. alopecuroides* leaves in Yili Valley ($p \leq 0.05$); they were positively correlated with C, N, and P contents and the N/P ratio, and negatively correlated with C/P and C/N ratios. AK is the dominant factor affecting the P content of *S. alopecuroides* leaves. Appropriate reduction of K fertilizer is conducive to restraining the spread of *S. alopecuroides*.

## ACKNOWLEDGEMENTS

We thank Saisai Zhang, Xia Yang, and Xiao Liu for their help with field investigation and sample collection.

### Funding
This study was supported by the Opening Fund of The State Key Laboratory of Desert and Oasis Ecology (Fund number: G2020-02-02) and the Tianshan Youth Program, a special talent program in Xinjiang Uygur Autonomous Region (Fund number: 2018Q076). The funders had no role in study design, data collection and analysis, decision to publish, or preparation of the manuscript.

### Grant Disclosures
The following grant information was disclosed by the authors:
The State Key Laboratory of Desert and Oasis Ecology: G2020-02-02.
The Tianshan Youth Program, a special talent program in Xinjiang Uygur Autonomous Region: 2018Q076.

### Competing Interests
The authors declare there are no competing interests.

### Author Contributions
- Yulu Zhang and Dong Cui conceived and designed the experiments, performed the experiments, analyzed the data, prepared figures and/or tables, authored or reviewed drafts of the paper, and approved the final draft.
- Yuhai Yang and Haijun Liu performed the experiments, authored or reviewed drafts of the paper, and approved the final draft.
- Haijun Yang and Yang Zhao performed the experiments, analyzed the data, authored or reviewed drafts of the paper, and approved the final draft.

### Data Availability
The original measurements are in the Supplemental File.

### Supplemental Information
Supplemental information for this article can be found online at http://dx.doi.org/10.7717/peerj.11701#supplemental-information.

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
