# Peer review of "Ecological stoichiometric characteristics and influencing factors of carbon, nitrogen, and phosphorus in the leaves of *Sophora alopecuroides* L. in the Yili River Valley, Xinjiang"

_PeerJ, doi:10.7717/peerj.11701_

## Round 0.1 · original submission · Major Revisions

This paper presented some new findings about the stoichiometry characteristics (C, N and P) in leaves of Sophora alopecuroides in west China. However, there are limitations of the introduction. A better literature review is needed to identify the knowledge gap in the research area. The discussion should also be improved, with a good explanation of the results. English needs thorough editing. The authors should present a language editing certificate in the cover letter when resubmitting the revised paper.

Reviewer 1 ·

Basic reporting

The manuscript #51868 submitted by Zhang and collaborators to Peer J is an original primary research contribution with a well-defined and relevant research question, with some not-so-well defined background and justifications to conduct the study.
The introduction needs to have a better background for the hypotheses. There are no assumptions in the introduction. In addition, the research significance of this article is not well explained. I am afraid that the introduction is not enough to make readers feel the value of research. In short, the introduction is poorly written and needs to be revised.
I am afraid the English used throughout was not professional. There are some grammatical and structural mistakes in the manuscript and I am not able to highlight/revise all of them. So, please revise the manuscript and improve it considerably in terms of grammatical and structural mistakes.

Experimental design

The experimental design appears to have followed a rigorous standard, I do not any concerns regarding the data collection.

Validity of the findings

All underlying data have been provided; they are robust, statistically sound.
But the discussion is confusing because it is based on losing hypotheses. Once the hypotheses are better formulated, the discussion should be easier to write.

Additional comments

1. For the study area map, please add the standard China map (complete borders and islands in the South China Sea). Many readers may not know the geographical location of Xinjiang.
2. The significance P-value in the article is the same as the abbreviation P for phosphorus. To avoid confusion, please revise the whole research. Such as Fig. 2.
3. How do the selected samples prove their representativeness? Please elaborate. It is necessary, otherwise, the conclusions cannot be supported.
4. What are the criteria for dividing the four types of plots? How are plant samples collected? The number of sample replicates? These are not explained clearly.
5. The experimental method is not clear. Please explain the test methods of C, N, P carefully, this is a very important step. For example, what instrument is used for N and P tests?
6. L114, “0~10、10~20、20~30”, Is there a "、" in English? There are too many such small mistakes, please carefully modify the full text.
7. L164, The unit of "470.09, 32.71," must be clearly written, don’t be lazy
8. L217, Have you also considered the K content of the soil? Why is this indicator not seen elsewhere in the article?
9. L268, I disagree with the conclusion of the sentence "The stoichiometric ratios of C, N and P in leaves of plants are relatively stable". C is relatively stable in plants, but N and P are not. Please read the relevant literature carefully.
10. Discussion writing is really bad. The thoughts of the discussion were too confusing, and the logic of the discussion was very unclear.
11. The local stoichiometric characteristics are usually compared with the global or domestic mean, which seems not mentioned in the study. For the discussion, didn't the first thing to compare Sophora alopecuroides with other research results?
12. Fig. 2 needs to be adjusted, the image resolution is low and unsightly; Table 3 does not indicate the level of significance; Fig. 4 Different font sizes.
In brief, the biggest problem with this article is language and logic. In addition, the introduction of the article and experimental methods are incomplete, and the discussion is even more unclear. Therefore, this article needs to be revised very seriously and rigorously. Otherwise, the article will be difficult to publish in the current state.

Reviewer 2 ·

Basic reporting

I recommend the authors to have a second look at the English editing of the manuscript.
Formatting of citations in text were incorrect.

Experimental design

More details should be added in the text

Validity of the findings

Part of conclusions looks like results, and draws no general conclusions.

Additional comments

1.Title:Ecological stoichiometry characteristics of carbon, nitrogen and phosphorus in leaves of Sophora alopecuroides in the Yili River Valley, Xinjiang
2.Abstract: authors should tell us what are the scientific issues in backgrounds,why? rather than "in order to....."
3.N and P contents should be concertrations
4.Authors must follow the rules of Latin names for plants in text.
5.The presentation of introduction is also not well-structured. why did authors conduct this experiment? what are the shortages of previously study? Authos should focus on the scientific issues. the current version is still insufficient and the logic is vague.
6.Three-layer soil samples were collected,Why? Where dose the majority of root distribution of plant? how the data of soil physical and chemical properties were used in the statistical analysis?
7.Sophora alopecuroides is a leguminous plants, nitrogen-fixing should be discussed and explain.
8.four habitats in the text are huge difference, such as: temperture, precipitation,fertilization in farmland,authors should explain in the manuscript.
9 Formatting of citations in text were incorrect.
10.Part of conclusions looks like results, and draws no general conclusions.
11.I recommend the authors to have a second look at the English editing of the manuscript.

Reviewer 3 ·

Basic reporting

no comment

Experimental design

The experimental design is problematic.

Validity of the findings

no comment

Additional comments

In recent decades, the C, N, P ecological stoichiometry characteristics of plant leaves at different time scales and spatial scales have been well studied. The authors study the differences in ecological stoichiometry characteristics of leaf carbon, nitrogen and phosphorus of Sophora alopecuroides in four habitats in the Yili River Valley and its influencing factors. This research is valuable, but there are many problems. The English presentation also needs careful polishing.

1. In the Abstract, the abbreviations of C, N, and P should be given in L29. Across the text, the Latin name of Sophora alopecuroides was not used standardizedly.
2. L46, that studies the changes rule? changes?
3. The content of influencing factors of plant leaf stoichiometry in the Introduction is too simple and should be described separately, and you should focus on your research object and focus on the discussion.
4. L72, what’s F resistance?
5. L83, carbon, nitrogen and phosphorus or C, N, and P?
6. A clear scientific question or hypothesis needs to be presented in the Introduction.
7. L93, 80°09′E—84°56′, 42°14′N—44°50′N?
8. L100, ℃? Or °C?
9. L114-115, a total of 12 soil samples and 12 plant samples was collected in 4 plots. Only 12 samples? It’s very small for stoichiometric analysis. And your repeat is only three, also small. In total, your experimental design is problematic.
10. L115-116, Leaves were dried at 105 OC for 24 h. Generally, the temperature ranged from 60-75 °C. Your temperature is too high.
11. L156. what is LGA?
12. L164, Yili River Valley wetland, wetland?
13. L213, factors of soil physical and chemical or soil physical-chemical factors?
14. L234-242, these explanations for CCA should be placed in “Statistical analysis”.
15. L274, table 2? Table 2?
16. Sophora alopecuroides is a nitrogen fixing plant species, so it’s generally N-limited; however, it might be also not P-limited, due to it’s higher leaf P content (mean=1.53 mg/g) than 1.0 mg/g. Therefore, the rule for judging restrictive elements, i.e., 14 or 16, is not effective for nitrogen fixing plants to a large extent.
17. The quality of Figure 1 is poor.
18. In Table 2, are you sure the differences of leaf P among four habitats are right?
19. Please also check the multiple comparison results in Table 3.
20. Check all Tables and Figures to avoid errors.

---

## Round 0.2 · Major Revisions

Following reviewers’ comments, the authors have made extensive revisions of the original version. However, more efforts are still needed to improve the writing. For the citation in text, only the author’s surname is needed. The reference list should be sorted in the alphabetical order of the authors’ names. More importantly, English needs thorough editing. The authors should present a language editing certificate in the cover letter when resubmitting the revised paper.

---

## Round 0.3 · Major Revisions

The authors have answered some questions and made the according changes of the manuscript. However, more proper answers to some questions are still needed.

#6, it is not just deleting one reference. What do you think about the C, N, and P.

#15, add serval research aims/objectives in the end of the Introduction.

L73, Ref. Xiaonan et al., 2004, is Xiaonan first name or surname? Also, this reference cannot be found in the reference list in the end.

Please carefully check all writing.

English needs thorough editing, including references. References are listed fully in alphabetical order according to the last name of the first author. The authors should present a language editing certificate in the cover letter when resubmitting the revised paper.

---

## Round 0.4 · Major Revisions

The authors have made some good changes. However, there are still some problems.

For the citation in the text, only a surname is needed. Correct Duan Xiaonan.

I cannot see your research aims. It is better to start with “The main research aims of the current study are …….”

Your reference list still has some problems, for example the orders of Cui et al. 2018, Cleland et al. 2011, Luo & Gong 2016…….You should use Endnote or other software to make the bibliography.

The English is still poor. The authors must present a language editing certificate in the cover letter when resubmitting the revised paper.

---

## Round 0.5 · Major Revisions

I can see you have made some changes, but it is still very easy to see some problems in the manuscript. First, I am afraid the English is still poor. You mentioned ‘a language editing certificate has been provided’, but I cannot find the certificate in your folder. Fundamentally, the English needs to be further polished by a native English speaker.
Second, I can easily spot errors of the Bibliography. A reference list should be ordered in alphabetical order by the names of authors. As I mentioned before, if you use Endnote or other reference software, these kinds of errors can be avoided much more easily.

---

## Round 0.6 · Minor Revisions

I have read your new version. Unfortunately, I can still find some English errors, for example, on Line 78. There are still obvious errors in your Reference list. The reference list must be ordered in alphabetical order of authors' names. For example (just for example, PLEASE check your list VERY CAREFULLY), it is incorrect to put Cui et al. 2018 before Cleland 2011. If authors still find it difficult to compile the reference list, please ask the English Editing Service Company to do proper editing (sort out the order of references).

---

## Round 0.7 · accepted · Accept

The authors have revised the manuscript appropriately following reviewers’ comments. The new version has reached the publication standard of our journal. Congratulations.